# The Predictive Ability of Wildlife Value Orientations for Mammal Management Varies with Species Conservation Status and Provenance

**Vasileios J. Kontsiotis, Archimidis Triantafyllidis, Stylianos Telidis, Ioanna Eleftheriadou and Vasilios Liordos ***





Department of Forest and Natural Environment Sciences, International Hellenic University, P.O. Box 172,
66100 Drama, Greece; vkontsiotis@for.ihu.gr (V.J.K.); dakoulis_93@yahoo.gr (A.T.); stelios3527@gmail.com (S.T.);
iwanna_eleftheriadou@hotmail.com (I.E.)
* Correspondence: liordos@for.ihu.gr

**Abstract:** Wildlife value orientations (WVOs) can predict consensus or controversy over wildlife-related issues and are therefore important for their successful management. We carried out on-site face-to-face interviews with Greek people (n = 2392) to study two basic WVOs, i.e., domination (prioritize human well-being over wildlife) and mutualism (wildlife has rights just as humans). Our sample was more mutualism-oriented than domination-oriented; however, domination was a better predictor of management acceptability than mutualism. WVOs were better predictors of the acceptability of lethal strategies (shooting, destruction at breeding sites, 11–36% of variance explained) relative to taking no action (9–18%) and non-lethal strategies (e.g., compensation, fencing, trapping, and relocating, 0–13%). In addition, the predictive ability of WVOs, mostly for accepting lethal strategies, increased with the increasing severity of the conflict (crop damage, attacking domestic animals, 11–29%; disease transmission, 17–36%) and depending on species conservation status and provenance (endangered native brown bear (*Ursus arctos*), 11–20%; common native red fox (*Vulpes vulpes*), 12–31%; common exotic coypu (*Myocastor coypus*), 17–36%). Managers should consider these findings for developing education and outreach programs, especially when they intend to raise support for lethal strategies. In doing so, they would be able to subsequently implement effective wildlife management plans.

**Keywords:** questionnaire survey; general public; value orientations; cognitive hierarchy; conflict management; common species; rare species; Eastern Mediterranean

## 1. Introduction

Value orientations are networks of basic beliefs organized around values and provide contextual meaning to those values in relation to a particular domain, such as wildlife [1,2]. They are part of the cognitive hierarchy of human behavior, comprising values, value orientations, attitudes/norms, and behavioral intentions [3–5], and mediate between general values and specific attitudes or norms [5]. Values are extensively shared by all members of a society, are part of one's identity, and are enduring throughout life. They also transcend situations and are unlikely to account for much of the variability in attitudes and specific behaviors. In contrast, value orientations give meaning to the more abstract values and can predict differences in attitudes and behaviors because their strength varies among individuals. Fulton et al. [3] developed a measurement instrument for basic beliefs concerning human–wildlife interactions based on the concept of value orientation. They identified two dimensions of wildlife value orientations (WVOs), the 'wildlife use' and the 'wildlife protection' dimension. Manfredo et al. [1] recognized domination and mutualism as the two basic WVOs (previously referred to as the protection/use dimension). Mutualism is composed of two belief dimensions—a caring for and a social affiliation with wildlife—while domination is composed of the hunting and use of wildlife belief dimensions. Individuals

with a domination WVO prioritize human well-being over wildlife and treat wildlife as a resource to be used for human benefit. Individuals with a mutualism WVO view wildlife as part of one's social community, deserving of rights and care like humans. Those with a domination WVO are more likely to accept management strategies that result in death or harm to wildlife, while mutualists are less likely to support strategies resulting in death or harm to wildlife [6–8].

A few studies have examined the predictive ability of mutualism and domination WVOs for the acceptability of management strategies. Sijtsma et al. [6] found that WVOs were the best predictors of the acceptability of lethal control to minimize the impacts of geese and deer on agricultural crops in the Netherlands. Jacobs et al. [7]) studied the predictive ability of WVOs for the acceptability of strategies for managing geese and deer in the Netherlands, varying in severity and under different conflict situations. They reported that the predictive ability of WVOs was largest for the acceptability of the most severe strategies (hunting), followed by the least severe strategies (doing nothing), and the intermediate strategies (shaking eggs or applying contraceptives). Glas et al [8] evaluated the predictive ability of WVOs for six meso-predator species (striped skunks (*Mephitis mephitis*), coyotes (*Canis latrans*), common raccoons (*Procyon lotor*), red foxes (*Vulpes vulpes*), American badgers (*Taxidea taxus*), river otters (*Lontra canadensis*)) in Indiana, USA, across conflict situations and management strategies varying in severity. They reported that WVOs explained most of the variance for lethal management actions (citizens hunt/trap, lethal removal by experts) and the least of it for trapping and relocating. Furthermore, they found that the predictive power of WVOs generally increased with the increasing severity of the situation, but without clear and consistent patterns. The predictive power of WVOs was also highest for the common coyotes and lowest for the uncommon river otters. Finally, all these studies found that domination was a better predictor of the acceptability of management strategies than mutualism.

Brown bears (*Ursus arctos*) are rare in Greece, declared as endangered by the Greek Red Data Book [9]. The two main brown bear populations are on the Pindos mountain range, the backbone of mainland Greece, running north to south from Albania to central Greece (estimated at about 300–350 individuals), and on the Rhodope mountain range in Northeast Greece (estimated at about 70–100 individuals), with the total Greek population estimated at about 500 individuals [10]. In Greece, the main impacts of the brown bear include livestock predation, mainly young cattle and single equines, and crop raiding, such as in small corn fields, vineyards, and apiaries [11]. Bear damage occurs throughout the year, but it is most common from May to October.

Outdoor livestock farming is extensively practiced in Greece where red foxes are known to attack domestic animals. Papageorgiou et al. [12] found goat kids, lambs, calves, and piglets in red fox diet. Red foxes are also responsible for transferring deadly diseases to people and their animals. In particular, rabies causes each year 55,000 deaths of people worldwide [13]. Greece had been rabies-free since 1987, with no human cases since 1970 [13]. From 2012 to 2013, rabies was diagnosed in 17 animals in north Greece, including 14 foxes, two dogs, and one cat [14].

Coypus (*Myocastor coypus*) are large semi-aquatic rodents native to southern South America [15], which have been introduced for fur farming into Europe, Asia, Africa, and North America [16,17]. Several coypu populations have become established along riverbanks and in wetlands in the areas where they were introduced, following their accidental and/or intentional release. Their high reproductive rates and their habit to consume whole plants, including roots, allows for their rapid population increase and spatial expansion and has severe impacts, predominantly damage to crops and disease transmission [18,19]. For these reasons, coypu is on the list of the 100 World's Worst Invasive Alien Species [20]. In Greece, coypus are present in rivers, lakes, and irrigation canals, mainly in the mainland, but also on islands (e.g., Corfu, Lefkada). The Greek coypu populations are escapees from fur farms in Greece and adjacent countries (i.e., Bulgaria and North Macedonia). Local authorities increasingly receive complaints from farmers,

mainly corn producers, whose fields are located near water bodies, but also fruit and vegetable producers, who also fear disease transmission from the rodent to their animals and ultimately to people through the food chain, mainly via fecal contamination of food and water (Directorate of Animal Husbandry Systems, Hellenic Ministry of Agricultural Development and Food, unpublished data).

The effectiveness of a management strategy at reducing impacts does not imply that conflicts are addressed, unless all interested parties support its use [21]. The predictive potential of WVOs for the acceptability of management strategies would help wildlife managers make informed decisions. Understanding the WVOs of a population would allow developing and implementing effective management plans, tailored to people's needs and expectations. Our study's aim was therefore to examine the predictive ability of WVOs for the acceptability of strategies for managing conflicts in the Greek population. Previous research has assessed the predictive ability of WVOs for management strategies differing in the degree of harm to wildlife, in conflict situations differing in severity, and for common and uncommon species [6–8]. We extended our previous research by examining the predictive ability of WVOs for the acceptability of management strategies for three mammal species differing in conservation status and provenance: the rare native brown bear, the common native red fox, and the common exotic coypu. Our research objectives were to assess the differences in the predictive ability of WVOs for the acceptability of management strategies between: (a) mammal species differing in conservation status and provenance, (b) management strategies differing in degree of harm to the species, and (c) conflict situations varying in severity.

Based on previous research and aims and objectives of our study, we hypothesized:

**Hypothesis (H1).** *The predictive ability of WVOs for the acceptability of management strategies will increase with their increasing harm to mammal species (i.e., taking no action, compensation, fencing versus destruction at breeding site, shooting);*

**Hypothesis (H2).** *The predictive ability of WVOs for the acceptability of management strategies will increase with increasing severity of the conflict (i.e., crop damage, predation on domestic animals versus disease transmission);*

**Hypothesis (H3).** *The predictive ability of WVOs for the acceptability of management strategies will be higher for common and exotic species than for rare and native species (i.e., brown bear versus red fox versus coypu);*

**Hypothesis (H4).** *Domination will be a better predictor of the acceptability of management strategies than mutualism.*

## 2. Materials and Methods

### 2.1. Sampling Protocol and Sample Size

The study was carried out in the 13 administrative Regions of Greece (Figure 1). Data were collected from on-site face-to-face surveys with adult Greek residents (aged 18–80), between March 2017 and September 2018. Cities, towns, and villages were visited in all Regions during open market hours (9.00–15.00 and 17.00–21.00, from Monday to Saturday). Every fifth person passing in front of the researcher was asked to participate by completing a questionnaire [22]. In cases in which more than five persons had passed while a questionnaire was being completed, the first person encountered upon completion was selected. It took respondents 15 min on average to complete the questionnaire.

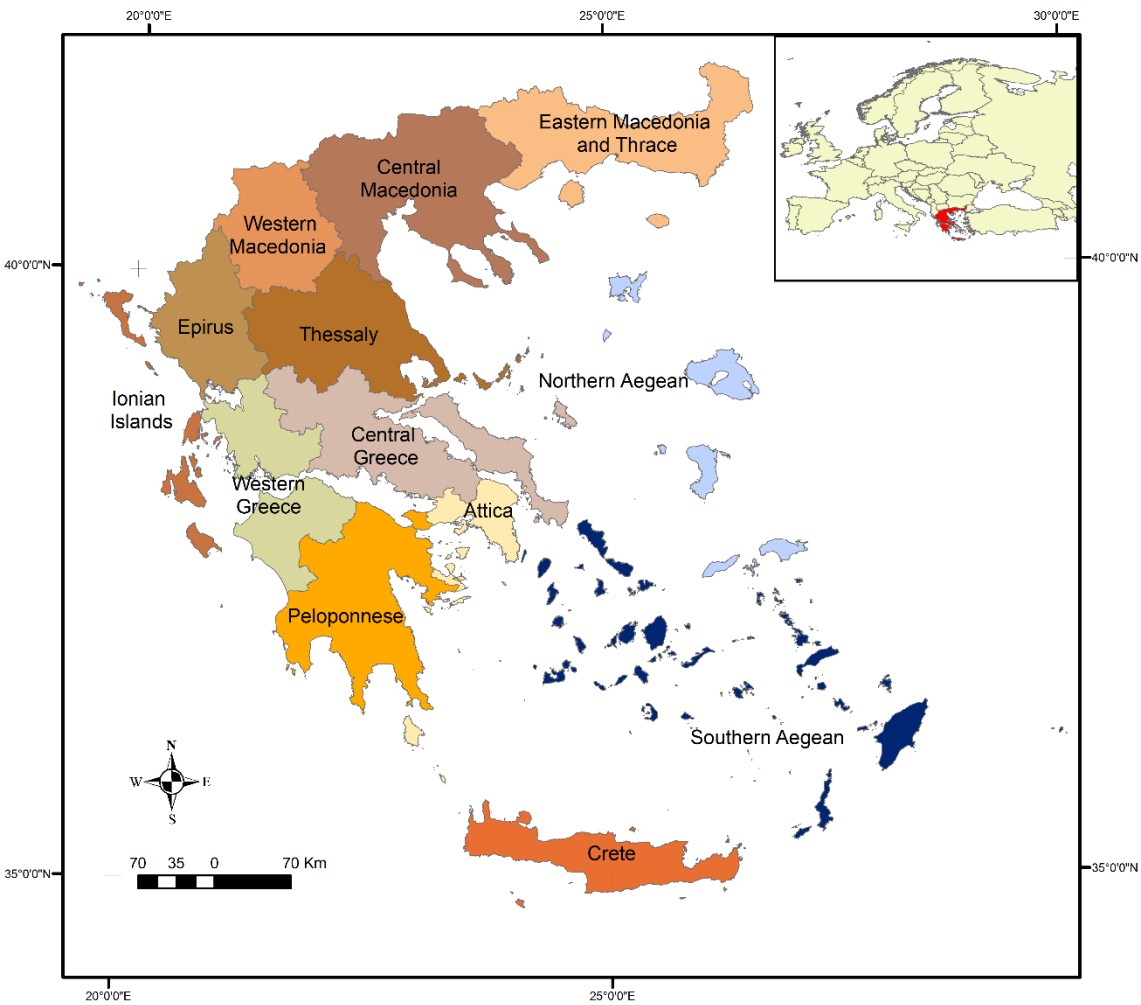

**Figure 1.** Map showing the 13 administrative regions of Greece, where the survey was carried out.

A total of 2392 questionnaires were completed, with 295 refusals, yielding a response rate of 89%. Our sample was adequate for Greece's population (10,730,000 people [23]) at a confidence level of 99% and a margin of error of 3% [22]. Greece's population has a 51.4% female/48.6% male gender ratio (50.8%/49.2% in this study); the age ratio, after excluding those under 18 and over 80, is 28.5%/37.1%/34.4% (32.4%/34.9%/32.6% in this study) for the 18–34-, 35–54-, and 55+-year-old age classes, respectively; the lower/higher educational level ratio is 73.4%/26.6% (70.9%/29.1% in this study) [23]; the rural/urban ratio is 21.0%/79.0% (23.7%/76.3% in this study) [24]. The sample's gender, age, educational level, and current residence (rural/urban) ratios were not different from those of the general population (gender: $\chi^2 = 0.064$, df = 1, $p = 0.769$; age: $\chi^2 = 4.481$, df = 2, $p = 0.106$; educational level: $\chi^2 = 1.790$, df = 1, $p = 0.166$; current residence: $\chi^2 = 2.554$, df = 1, $p = 0.099$) (see Liordos et al. [25] for further information).

*2.2. Survey Design*

In the first part of the questionnaire, we investigated two basic WVOs, assessed by 19 statements in total, following Jacobs et al. [7]. The exact wording of statements is given in Table 2 and Table S1. The domination value orientation (10 statements) was based on two basic belief dimensions (appropriate use beliefs (six statements), hunting beliefs (four statements)). The mutualism value orientation (nine statements) contained two basic beliefs (social affiliation beliefs (four statements), caring beliefs (five statements)). All variables were coded on seven-point scales ranging from 1, "strongly disagree", to 7, "strongly agree".

The second part of the questionnaire contained six conflict scenarios, involving species with different conservation and provenance status: (1) brown bears damage crops; (2) brown bears attack domestic animals; (3) red foxes attack domestic animals; (4) red foxes transfer disease; (5) coypus damage crops; (6) coypus transfer disease. Five management strategies were offered under scenarios (1) and (2), varying in degree of harm to wildlife: (1) take no action; (2) award suitable compensation; (3) use fencing for the protection of crops or livestock; (4) trap and relocate nuisance animals; (5) reduce populations through shooting carried out by state experts. 'Destruction at breeding sites' replaced 'trapping and relocation' in scenarios (3), (4), (5), and (6). 'Vaccination' replaced 'fencing' in scenario (4). 'Provision of alternative food' replaced 'fencing' in scenarios (5) and (6). Survey questions were structured as, using scenario 1 as an example, "Brown bears are threatened animals that visit crops to feed. When they cause significant damage, how acceptable or unacceptable would be for you to: (1) take no action; (2) award suitable compensation; (3) use fencing for the protection of crops; (4) trap and relocate nuisance animals; (5) reduce their populations through shooting carried out by state experts." Participants were then asked to assess each management strategy as acceptable (1) or unacceptable (0).

*2.3. Data Analysis*

Reliability and confirmatory factor analysis were used to validate WVO theoretical constructs. Model fit was assessed using five indicators: $\chi^2/\mathrm{df}$, with acceptable value $\leq 3$; comparative-fit index, CFI $\geq 0.95$; goodness-of-fit index, GFI $\geq 0.90$; normed-fit index, NFI $\geq 0.95$; root mean-square residual, RMR $\leq 0.08$ [26]. Cronbach's alpha was used to examine the reliability of each multi-item scale, with values greater than 0.70 considered acceptable [27]. Responses concerning the acceptability of management strategies were coded as acceptable = 1 and unacceptable = 0. Binary logistic regression was used for evaluating the predictive potential of mutualism and domination WVOs for the acceptability of each management strategy [7]. Logistic regression and reliability analysis were performed with SPSS Statistics, and confirmatory factor analysis with SPSS Amos statistical software (version 21.0, IBM Corp., Armonk, NY, USA, 2012). The significance level was set at $\alpha = 0.05$.

## 3. Results

Confirmatory factor analysis provided a good fit for the data ($\chi^2/\mathrm{df}$ = 2.87, CFI = 0.98, GFI = 0.93, NFI = 0.96, RMR = 0.053) and supported the constructs associated with the latent variables, with standardized factor loadings being statistically significant at $p < 0.001$ and above the minimum criterion of 0.40 used to denote practical significance (Table 1). In addition, the internal reliability of the domination (Cronbach's alpha = 0.80) and mutualism (Cronbach's alpha = 0.86) WVOs was high. Deleting any item from their basic belief dimension did not improve reliability. Therefore, composite indices were created for domination and mutualism. These indices were used for further analyses. On average, the respondents were more mutualism-oriented (mean 4.64 ± 1.45 SD) than domination-oriented (mean 3.31 ± 1.61) (see Liordos et al. [25] for further analyses).

**Table 1.** Reliability and confirmatory factor analysis (CFA) of wildlife value orientation statements.

| Wildlife Value Orientation Statements [a] | CFA | Reliability Analysis | | |
|---|---|---|---|---|
| | Factor Loadings [c] | Item Total Correlation | Alpha if Item Deleted | Cronbach's Alpha |
| *Domination* | | | | 0.80 |
| *Appropriate use beliefs* | | | | 0.76 |
| Humans should manage fish and wildlife populations so that humans benefit. | 0.68 | 0.42 | 0.71 | |
| The needs of humans should take priority over fish and wildlife protection. | 0.76 | 0.50 | 0.69 | |
| It is acceptable for people to kill wildlife if they think it poses a threat to their life. | 0.60 | 0.48 | 0.69 | |
| It is acceptable for people to kill wildlife if they think it poses a threat to their property. | 0.71 | 0.59 | 0.66 | |
| It is acceptable to use fish and wildlife in research even if it may harm or kill some animals. | 0.65 | 0.45 | 0.7 | |
| Fish and wildlife are on earth primarily for people to use. | 0.90 | 0.38 | 0.72 | |
| *Hunting beliefs* | | | | 0.75 |
| We should strive for a world where there is an abundance of fish and wildlife for hunting and fishing. | 0.66 | 0.43 | 0.68 | |

**Table 1.** *Cont.*

| Wildlife Value Orientation Statements [a] | CFA | Reliability Analysis | | |
|---|---|---|---|---|
| | Factor Loadings [c] | Item Total Correlation | Alpha if Item Deleted | Cronbach's Alpha |
| Hunting is cruel and inhumane to the animals. [b] | 0.54 | 0.57 | 0.52 | |
| Hunting does not respect the lives of animals. [b] | 0.59 | 0.59 | 0.51 | |
| People who want to hunt should be provided the opportunity to do so. | 0.69 | 0.42 | 0.68 | |
| *Mutualism* | | | | 0.86 |
| *Social affiliation beliefs* | | | | 0.77 |
| We should strive for a world where humans and fish and wildlife can live side by side without fear. | 0.55 | 0.46 | 0.74 | |
| I view all living things as part of one big family. | 0.66 | 0.60 | 0.66 | |
| Animals should have rights similar to the rights of humans. | 0.82 | 0.53 | 0.70 | |
| Wildlife is like my family and I want to protect it. | 0.81 | 0.61 | 0.66 | |
| *Caring beliefs* | | | | 0.81 |
| I care about animals as much as I do for people. | 0.77 | 0.48 | 0.79 | |
| It would be more rewarding to me to help animals rather than people. | 0.51 | 0.43 | 0.79 | |
| I take great comfort in the relationships I have with animals. | 0.69 | 0.68 | 0.73 | |
| I feel a strong emotional bond with animals. | 0.82 | 0.72 | 0.71 | |
| I value the sense of companionship I receive from animals. | 0.75 | 0.62 | 0.75 | |

[a] Variables coded on seven-point scales ranging from 1 (Strongly disagree) to 7 (Strongly agree); [b] Item was reverse-coded prior to analysis; [c] All *t* values for standardized factor loadings were significant at $p < 0.001$.

Across scenarios, taking no action was highly unacceptable (acceptability 11–23%), and harmful management strategies were also highly unacceptable (destruction at breeding sites and shooting) in the least severe scenarios (crop damage and domestic animal predation: acceptability 7–21%), although more controversial in the more severe scenario (disease transmission: acceptability 17–36%) (Table 2). In contrast, management strategies that are not harmful to wildlife were the most acceptable (compensation, fencing, vaccination, provision of alternative food: acceptability 44–86%) except for compensation in the disease transmission scenario (acceptability 13–18%).

**Table 2.** Logistic regression models of wildlife value orientation dimensions predicting the acceptability of wildlife management strategies for different species and situations.

| | Acceptable (%) | Unacceptable (%) | Odds Ratio Domination | Odds Ratio Mutualism | Nagelkerke $R^2$ |
|---|---|---|---|---|---|
| *Brown bears damage crops* | | | | | |
| No action | 23.1 | 76.9 | 0.76 ** | 1.31 * | 0.14 |
| Compensation | 60.4 | 39.6 | 1.21 | 1.27 * | 0.04 |
| Fencing | 84.4 | 15.6 | 1.19 | 1.17 | 0.02 |
| Trapping and relocation | 38.6 | 61.4 | 1.41 ** | 0.91 | 0.11 |
| Shooting | 7.4 | 92.6 | 1.28 * | 0.80 * | 0.11 |
| *Brown bears attack domestic animals* | | | | | |
| No action | 21.9 | 78.1 | 0.65 *** | 1.29 * | 0.14 |
| Compensation | 60.7 | 39.3 | 1.28 * | 1.27 * | 0.08 |
| Fencing | 84.6 | 15.4 | 1.11 | 1.21 | 0.02 |
| Trapping and relocation | 41.0 | 59.0 | 1.65 *** | 1.07 | 0.13 |
| Shooting | 10.0 | 90.0 | 1.68 *** | 0.83 * | 0.20 |
| *Red foxes attack domestic animals* | | | | | |
| No action | 18.1 | 81.9 | 0.83 * | 1.25 * | 0.09 |
| Compensation | 44.1 | 55.9 | 1.27 * | 1.31 * | 0.08 |
| Fencing | 80.6 | 19.4 | 1.10 | 1.20 | 0.02 |
| Destruction at breeding sites | 20.1 | 79.9 | 1.57 ** | 0.99 | 0.12 |
| Shooting | 16.9 | 83.1 | 1.88 *** | 0.78 * | 0.29 |
| *Red foxes transmit disease* | | | | | |
| No action | 12.4 | 87.6 | 0.61 *** | 1.07 | 0.15 |
| Compensation | 18.3 | 81.7 | 0.87 | 1.02 | 0.01 |
| Vaccination | 85.4 | 14.6 | 1.11 | 1.18 | 0.02 |
| Destruction at breeding sites | 17.4 | 82.6 | 2.12 *** | 0.97 | 0.19 |
| Shooting | 25.3 | 74.7 | 2.86 *** | 0.89 | 0.31 |
| *Coypus damage crops* | | | | | |
| No action | 14.4 | 85.6 | 0.76 ** | 1.02 | 0.12 |
| Compensation | 54.5 | 45.5 | 1.29 * | 1.27 * | 0.08 |
| Provision of alternative food | 78.3 | 21.7 | 0.94 | 1.02 | 0.00 |
| Destruction at breeding sites | 20.7 | 79.3 | 1.76 *** | 1.14 | 0.17 |
| Shooting | 19.1 | 80.9 | 2.37 *** | 1.10 | 0.25 |
| *Coypus transmit disease* | | | | | |
| No action | 10.5 | 89.5 | 0.50 *** | 0.88 | 0.18 |
| Compensation | 12.7 | 87.3 | 0.85 | 0.99 | 0.01 |
| Provision of alternative food | 68.7 | 31.3 | 0.91 | 1.11 | 0.01 |
| Destruction at breeding sites | 36.3 | 63.7 | 2.51 *** | 1.17 | 0.28 |
| Shooting | 33.1 | 66.9 | 3.46 *** | 1.23 | 0.36 |

* $p < 0.05$; ** $p < 0.01$; *** $p < 0.001$.

Across species and scenarios, the predictive ability of WVOs was consistently larger for the acceptability of strategies that are most harmful to wildlife (Table 2). WVOs explained 11–36% of the variance in shooting and 12–28% of it in destruction at breeding sites. WVOs explained smaller amount of the variance in taking no action (9–18%) and trapping and relocating (11–13%), while strategies that are not harmful to wildlife explained the smallest amounts of the variance (0–8%). The predictive ability of WVOs for the most harmful strategies also increased with the increasing severity of the scenarios. Percentages of the explained variance varied between 11 and 29% in the crop damage and attacking domestic animals scenarios and between 17 and 36% in the disease transmission scenarios for the shooting and destruction at breeding sites management strategies.

Across species, the predictive ability of WVOs varied with species conservation status and provenance (Table 2). WVOs explained larger amounts of the variance in the taking no action strategy for the endangered native brown bear (14%) and the common native red fox (9–15%) than for the common exotic coypu (12–18%). In contrast, differences were reversed for the most harmful management strategies. WVOs explained smaller amounts of the variance in shooting and/or destruction at breeding sites strategies for the endangered native brown bear (11–20%) than for the common native red fox (12–31%) and the common exotic coypu (17–36%).

Finally, domination was a better predictor than mutualism across scenarios and management strategies (Table 2). Most importantly, the odds ratio for the acceptability of shooting varied between 1.28 and 3.46 for domination and between 0.78 and 1.23 for mutualism.

## 4. Discussion

### 4.1. The Predictive Ability of WVOs

In agreement with previous similar studies, WVOs were significant predictors of the acceptability of strategies for managing mammal species [6–8]. In all scenarios, the predictive ability of WVOs was consistently larger for the acceptability of strategies that are lethal to wildlife (i.e., shooting, destruction at breeding sites) and smaller for strategies that are not harmful to wildlife (i.e., compensation, fencing, vaccination, provision of alternative food). Previous studies also reported similar trends [6–8]. The low predictive ability of WVOs for strategies that are not harmful to wildlife suggested a low potential for conflict among and within individuals. In contrast, the large predictability of WVOs for lethal strategies, which also increased with the increasing severity of the management situation, suggested a high potential for conflict among and within individuals. The public usually reaches consensus for the application of non-harmful strategies for managing mammal species [28–35]. In contrast, the more invasive, lethal strategies are most often highly controversial among people, more so in the most severe management situations [28–35]. A conflict also involves an internal, within an individual, collision between values. An individual holds various values [36], and one's behavior is most often guided by more than one value [5]. Let us suppose that a person holds the values of safety and respect of life. In the context of wildlife management, there is a high internal collision of these values when a red fox transfers deadly disease to people. In this example, values collide; one must weigh people's safety against the respect for the life of the red fox. To resolve the internal collision, one value must assume priority over the other values [5,36]. WVOs resolve internal collisions by prioritizing certain values over other values in the context of wildlife [5]. Jacobs et al. [7] reported findings similar to ours and concluded that the predictive ability of WVOs increases as the potential internal collision of values increases, and that the potential internal collision of values increases as a function of the severity of the management strategy for wildlife (e.g., fencing versus shooting) and the severity the conflict situation (e.g., crop damage versus disease transmission).

Irrespective of the conflict situation, WVOs were better predictors of lethal strategies for the coypu than for the red fox and the brown bear. These findings imply a higher potential for conflict for a common exotic than for a common native and, especially, a rare native

species, both between and within individuals. Endangered brown bears are charismatic species, conservation icons that people find attractive and likeable, although fearful [37,38]. Red foxes are also perceived as attractive animals, and people enjoy watching them [39]. In contrast, people see coypus as "disgusting big rats", intruders that threaten their property and fear disease transmission from the rodent to domestic animals and ultimately to people through the food chain, mainly via fecal contamination of food and water [40]. Indeed, in Greece, people behave differently towards these species. They, mostly farmers, resort to fencing for protecting their property from brown bears and red foxes, demanding compensation when damage occurs [11,32]. They also ask for relocating individual animals that have been habituated to the human presence and display aggressive behavior. In contrast, farmers actively pursuit and illegally kill coypus when they find them on their land, fearing losing their crops [30]. Previous research has shown that lethal strategies were more acceptable and controversial for managing coypus than native pest species [28]. It also seemed that people assigned relative priorities to the conservation of species and, hence, to the respect of their life: first priority to the charismatic, rare, and native brown bear, second priority to the charismatic, common red fox, and third priority to the uncharismatic, common, and exotic coypu. Giving priority to the respect of life over impact management helped reach a decision and thus eased the internal collision of values and lowered the predicted ability of WVOs for brown bear management situations, as compared to those for red foxes and especially coypus.

Our results suggest, in agreement with theoretical predictions [1,4,5,41] and previous research [6–8], that people with a domination orientation were more likely to accept strategies that imply harm to animals than people with a mutualism orientation. Domination was also a better predictor than mutualism across scenarios and management strategies. Previous research also suggested that domination had higher predictive ability than mutualism in management situations involving both common and uncommon species [6–8]. In contrast, Hermann et al. [42] reported that mutualism was a better predictor of a conservation intervention, supporting the return of the endangered grey wolf (*Canis lupus*) and European bison (*Bison bonasus*) in Germany. The level of endangerment of a species is an important predictor of the support for wildlife conservation [37,38,43,44]. In addition, people place higher values on species when they are rare and a lower value on species perceived as common [45]. Based on such findings, Hermann et al. [42] and Jacobs et al. [7] suggested that mutualism might better predict support for the conservation of rare species, while domination might better explain support for the control of common species in common situations. The brown bear, similarly to the gray wolf and the European bison, is a rare, endangered species, perceived by the public as attractive and worthy of protection [37,38]. However, we found, in contrast to the findings and predictions of Hermann et al. [42] and Jacobs et al. [7], that domination was consistently a better predictor of the acceptability of management strategies than mutualism. Another interesting finding from our study was that the predictive ability for the acceptability of management strategies of domination was lower and that of mutualism was higher in all the conflict situations that involved a rare native species (brown bear) than in situations involving a common native (red fox) and, more importantly, a common exotic species (coypu). Based on our findings and on findings from previous research [6–8], we extend the arguments of Hermann et al. [42] and Jacobs et al. [7] by further arguing that the interplay between situational context (e.g., conservation management versus conflict management), species conservation status (i.e., rare or common), and species provenance (native or exotic) determines which WVO, domination or mutualism, would be a better predictor of wildlife management interventions and at what intensity. However, additional research is needed to further confirm such patterns.

*4.2. Management Implications*

Research has shown that management plans and strategies incorporating WVOs are more likely to be accepted by the public and, therefore, more effective [46–48]. The high pre-

dictive ability of WVOs in our study suggests that WVOs could help managers to measure support for management strategies and make informed management decisions [49]. WVOs had higher predictive value for lethal strategies and lower predictive value for strategies that do not cause harm to wildlife. This means that the internal conflict of values is higher for lethal than for non-lethal strategies. As a result, non-lethal strategies are more easily accepted by the public, while lethal strategies are more controversial. Therefore, managers should preferably choose non-lethal strategies among these that are deemed effective for managing a situation. Compensation, fencing, and trapping and relocation of nuisance individuals have been found effective for reducing the damage of human property by brown bears [11]. Compensation and fencing have been found effective for reducing the damage of human property by red foxes [50]. Vaccination is considered the most effective strategy for eliminating rabies from red foxes and preventing the spread of disease to domestic animals (predominantly dogs) and humans [13,14]. Managers should choose the most effective among these non-lethal strategies, or a combination of them, for successfully implementing wildlife impact management. In contrast, complete eradication is considered as the most effective strategy for managing the invasive coypu, because of its behavioral characteristics (high reproductive outcome, fast linear expansion across waterbodies due to its habit to consume whole plants, including roots) [16–19,51]. As the internal conflict for lethal strategies such as shooting and destruction at breeding sites appeared high, managers should develop communication and outreach plans to raise public support for the most suitable management strategies. Such plans should consider the WVOs of target groups and be tailored to the needs and expectations of these groups. However, as WVOs are relatively stable [1,5,52] and the internal conflict of values is intense in lethal strategies, managers should keep in mind that communication and outreach plans are less likely to be successful for lethal than for non-lethal strategies [7].

## 5. Conclusions

We measured the domination and mutualism WVOs of the Greek population and examined their predictive ability for management strategies in several conflict situations. The Greek public was more mutualism-oriented than domination-oriented; however, domination was a better predictor of conflict management acceptance than mutualism. The predictive ability of WVOs was generally: (a) higher for non-lethal than for lethal strategies, (b) higher in the more severe (i.e., disease transmission) than in the less severe conflict situations (i.e., crop raiding, attacking domestic animals), (c) higher for common native or exotic species than for rare native species. These findings would allow managers to develop and implement effective, species- and situation-specific, management plans by incorporating the WVOs of public groups. Further research should be carried out to confirm the findings of this and previous studies [6–8,42], regarding the predictive ability of WVOs for conservation and management interventions in relation to intervention and situation severity as well as species conservation status and provenance. In this study, we were interested in examining the survey participants' preferences for wildlife management. We did not provide them with information about the cost, effectiveness, technical feasibility, and availability of the management strategies, neither did we assessed their knowledge about these issues. Future research should also address these issues, because they would help managers design targeted education and outreach programs aiming at reaching consensus for the most suitable among the available management strategies.

**Supplementary Materials:** The following are available online at https://www.mdpi.com/article/10 .3390/su132011335/s1, Table S1: The survey instrument.

**Author Contributions:** Conceptualization, V.L. and V.J.K.; investigation, I.E.; methodology, V.L., V.J.K., I.E., S.T. and A.T.; software, V.L.; validation, V.L. and V.J.K.; formal analysis, V.L. and V.J.K.; resources, V.L., V.J.K., I.E., S.T. and A.T.; data curation, V.L. and V.J.K.; writing–original draft preparation.; V.L.; writing–reviewing and editing, V.L., V.J.K., I.E., S.T. and A.T.; visualization, V.L.;

supervision, V.L. and V.J.K.; project administration, V.L. All authors have read and agreed to the published version of the manuscript.

**Funding:** This research received no external funding.

**Institutional Review Board Statement:** The study was conducted according to the guidelines of the Declaration of Helsinki and adhered to the ethical standards laid out by the Research and Academic Committee of the International Hellenic University.

**Informed Consent Statement:** We sought informed consent from all the participants and maintained anonymity at all stages of the research.

**Data Availability Statement:** The data presented in this study are available on reasonable request from the corresponding author.

**Acknowledgments:** We thank the survey participants for sharing their time and opinion with us. We also thank three anonymous reviewers whose comments and suggestions helped greatly improve the manuscript.

**Conflicts of Interest:** The authors declare no conflict of interest.

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
