# Peer review of "The Predictive Ability of Wildlife Value Orientations for Mammal Management Varies with Species Conservation Status and Provenance"

_sustainability, doi:10.3390/su132011335_

Round 1
Reviewer 1 Report
The article The Predictive Ability of Wildlife Value Orientations for 2
Mammal Management Varies with Species Conservation Status 3
and Provenance seems a very useful work for the management and conservation of wildlife. Generally its well written, but still needs language edits. however some specific suggestions for authors are given below:
Abstract: The abstract is very lengthy and dose not comply with the journal format. it is suggested to to squeeze the abstract as per journal format (200 words), for easy understandings for readers.
Line 13-14: Rephrase to "We carried out on-site face to interview (n=2,392) representative of Greek population.
Line 21: Change "nest sites" to "breeding sites"
Line 42: delete "and behaviors"
Line 50-51: change wildlife 'use' to 'wildlife use' and wildlife 'protection' to 'wildlife protection'
Line 70: mid-size to meso
Line 70-72: scientific names only should be in brackets, please edit throughout manuscript.
Line 81-82: "having been assigned to endangered
conservation status" to "declared as endangered by..."
Line 85: "totaling 500 individuals" will be much more elaborative if authors present the population of each site.
Line 85-6: Rephrase the sentence, attacks on live stock> livestock predation, equids>equines, feed on crops> crop raiding.
Line 80-91: practiced in Greece and, as red foxes to practiced in Greece where red foxes
Line 102: most importantly to predominantly
Line125: nonnative to exotic
Line 130: change to "Based on previous research and aims and objectives of our study we hypothesized:"
Materials and methods
How much the sample constitutes of total Greece population. please mention.
Please must provide the questionnaire in supplementary materials.
The authors have used Logistic regression model, how they selected best fir model, please explain, if based on AIC ? then mention and explain.
Discussion:
Line 244: nest sites to breeding sites,,,, check throughout manuscript
Line 268: implied a higher ... what?? please check it
Line 270-74: the respondents attitudes are more positive toward brown bear and red fox as compared to coypus, although the first two are also involved in livestock predation, crops raiding and diseases transmission. would the authors explain and add some solid facts for such kind of attitude from locals?
line 299: gray wolf to grey wolf
Line 348: Please elaborate the scale of severity of conflicts situations. which clearly reflects more and less severe situations.
Reviewer 2 Report
The article is an interesting human dimensions, survey-based analysis of the role of human values in support for a variety of theoretical wildlife management techniques. The sample size is high with a high rate of survey response from a human population in Greece. More detailed analysis of demographics (e.g. age, sex and rural vs urban) was not provided or discussed and could potentially be added as supplemental information or at least mentioned if there were no differences. As wildlife managers would likely also include other factors (e.g. cost, effectiveness and logistics) in decision making it is worth addressing that the survey had limitations of addressing this as it was assumed that the techniques (e.g. shooting, fencing, vaccines) were technically possible and available and did not address any potential differences in cost.
Some sections would benefit from minor editing.
Lines 13-18 defining the value categories is a very long sentence and should likely be split up into multiple sentences.
Lines 18-19 non-lethal, mutualism-oriented and domination-oriented should be hyphenated
Line 18 ;however should be prefaced by a semi-colon
Line 140: Formatting seems off for hypothesis 4
Line 161-161- square brackets should be used for terms within round brackets
Line 327-328 "proved effective" is awkward wording, please reword.
Line 328-329: Do you mean control rabies in foxes? I am not aware that rabies has been eliminated successfully in wild fox populations and do you mean controlling rabies within the fox population, preventing spread to domestic dogs, preventing human rabies or all three? This should be clarified.
Overall it is an excellent study with implications for wildlife management decision making.
Reviewer 3 Report
This study presents an interesting and original analysis of the predictive ability of WVOs and has the potential to provide useful background information to wildlife managers and decision makers.
The aims and objectives are clearly defined and overall the paper is well written. The data are well analysed, statistical analysis is straightforward and conclusions drawn reflect the findings. Therefore I believe this paper will make a good contribution to the journal.
I believe the manuscript needs minor revision and I would like to make few recommendations to the author(s) in order to improve this paper:
Methods
Section 2.1. Sampling protocol and sample size
Line 153-154: I am surprised that the sampling method that was employed (every fifth person passing in front of the researcher), which has a significant element of randomness (systematic random sample), has provided such an accurate representation of the gender, age, educational level and urban/rural structure of the overall Greek population. Unfortunately not enough information is provided in the paper as the authors simply cite the work of Liordos et al. I would very much like to see specific numbers regarding gender, age, educational level and urban/rural structure and how these are compared to national distribution. Alternatively the authors should remove the sample was not different to that of the population’s
Section 2.2. Survey design
Line 160: It is not clear why the authors are using a reference at the end of the sentence. DO they imply that they have used the same WVOs and statements as Jacobs et al 2014? If yes, then it should be stated clearly
Lines 159-162: Please indicate that the statements are available in Tables to follow.
It would be appropriate to present the statements. The nature of the statement may affect the people’s response. At the moment the reviewer cannot judge the appropriateness and the potential implications of the statements.
Round 2
Reviewer 1 Report
The authors have satisfactorily revised the MS as per raised concerns. Yet still i advice the authors to check for some English edits before the article is published.